# Effect of government revenue on economic growth of sub-Saharan Africa: Does institutional quality matter?

**Isubalew Daba Ayana** [1]*, **Wondaferahu Mulugeta Demissie**[2], **Atnafu Gebremeskel Sore**[3]

**1** Wollega University and Department of Economics, Research Scholar, Nekemte, Ethiopia, **2** Associate Professor of Economics, Ethiopian Civil Service University, Addis Ababa, Ethiopia, **3** Assistant Professor of Economics, and Addis Ababa University, Addis Ababa, Ethiopia

* isubalewd@wollegauniversity.edu.et

**Data Availability Statement:** All data files are available from the World Bank data base(https://databank.worldbank.org/reports.aspx?source=World-Development-Indicators) and Heritage

## Abstract

Following the approval of sustainable development goals at the global level, the link between fiscal policy, institutional quality, and economic growth has attracted special attention in economic literature. This study scrutinizes the effect of government revenue-institutional quality interaction on the economic growth of 43 Sub-Saharan Africa countries for the period of 2012–2022. Methodology-wise, the study employed the System Generalized Method of Moment (SGMM) to analyze the panel data gained from dependable data sources; the World Development Indicator and the Heritage Economic Freedom Index. The novelty of this study emanates from the estimation technique designated and the introduction of revenue-institutional quality into the economic growth model of SSA. The result of the study reveals that government revenue adversely affects economic growth while institutional quality positively enhances economic growth before interacting with each other. However, the interactive coefficient of government revenue and economic growth positively affected the real GDP growth rate of SSA countries over the study periods. Precisely, before interacting with institutional quality, a percentage change in government revenue, keeping all other things constant, leads to a 0.0866 percent decline in economic growth while it marks a 0.2329 percent upsurge in economic growth in the presence of institutional quality. The result of the study further shows that government revenue promotes the economic growth of the region when combined with institutional quality. On the other hand, foreign direct investment and openness to trade were the key sources of economic growth whereas the population growth rate adversely impacted economic growth in SSA countries. The policy implication of the study is that SSA needs to strengthen government revenue management. Further, the finding of the study implies that SSA countries need to improve institutional quality through promoting efficiency of the regulatory quality and the size of the SSA governments. In addition to this, the fast real GDP growth rate of SSA countries demands improved institutional quality indicators such as the rule of law and extended access to the open market.

economic foundation index(https://www.heritage.org/index/download/).

**Funding:** The author(s) received no specific funding for this work.

**Competing interests:** The authors have declared that no competing interests exist.

## 1. Introduction

Governments of countries across the globe shoulder enormous responsibilities which are prominently influenced by the level of government revenues. The governments require various amounts of resources to execute their social and economic wellbeing functions. The entire resource utilized by the government in running government jobs is dubbed as government revenue. It involves taxes, and revenue from administrative activities like grants, fines, and penalties [1].

Since the adoption of sustainable development goals in 2015, a new intuition into economic growth emerged from the angle of factors affecting economic growth. The effect of endogenous variables such as institutional quality was given huge attention. In Africa, the link between fiscal policy variables, institutional quality, and economic growth become the focus of policy makers following agenda 2063. Consequently, several empirical studies linked intuitional quality and growth in SSA countries and reported a positive relationship [2,3].

The government revenue can also be broadly classified as tax and non-tax revenue However, despite the various sources of revenue available, governments in SSA countries mainly relay on tax to operate their activities. This is because tax is a compulsory payment to the government of SSA like it is elsewhere in the world. Further, as individuals and companies must pay taxes, a government tends to depend on tax revenues. Thus, tax revenue is a source of revenue on which governments of SSA countries want to rely [4].

Following the tendency of SSA country's governments to rely on tax revenue, the effect of government revenue on economic growth has attracted the attention of policymakers and scientific investigators in the area. Unlike the ancient times when the tax was only to generate income for the government, tax revenue, and government revenue, in general, is one of the essential tools of fiscal policy. It is one of the instruments for achieving macroeconomic objectives such as controlling the level of inflation and unemployment [5]. Furthermore, government revenue is considered a tool for enhancing social justice and poverty alleviation [6].

Currently, the literature debate has surpassed a mere linkage between government revenue and the economic growth of countries. The point of argument that attracted economic literature and academic research during the second decade of the 21st century is the interaction of government revenue with the quality of institutions. The debate is very hot in SSA Africa, where the quality of institutional quality is questionable. For instance [7], who recently examined the effect of tax revenue and economic growth, found an endogenous linkage between tax revenue, institutional quality, and economic growth in the short run for lower and middle-income countries. Contrary to this [8], who applied a new econometric estimation approach, the frequency domain approach, demonstrated that there is no causal relationship between tax revenue and economic growth from 1980 to 2016.

The other empirical study that linked tax revenue, institutional quality, and economic growth is [9], which suggested better institutional quality for better revenue performance and in turn faster economic growth. By using a panel dataset of 95 developing countries and a two-system GMM approach from 1981 to 2015, their study found that tax revenue enhances economic growth. Furthermore, their study uncovered that tax revenue is highly associated with the level of tax performance which demands better institutional quality.

Similarly [10], linked taxation with intergenerational welfare as a means of boosting economic growth while [11] viewed government revenue from the angle of being competent in an internal market through better openness to trade [12]. Demonstrated that institutional quality is among the factors that determine tax revenue and in turn economic growth.

Despite many recent empirical works on the link between government revenue, institutional quality, and economic growth, the literature on SSA remains limited. Hence, the

objective of this study is to examine the effects of government revenue and its interaction with institutional quality on economic growth on the economic growth of SSA countries over the study periods of 2012–2022.

This paper has novel contributions in the following three ways. Firstly, this paper develops conceptual arguments exploring the relationship between government revenue, institutional quality, and economic growth across SSA countries. As it introduces the interaction between government revenue and institutional quality in the economic growth model, the paper explores whether the quality of institutions contributes or not to the economic growth of SSA countries.

Secondly, the paper uses a recent and larger dataset of 43 SSA countries for the past decade 2012–2022, and comprehensive indicators of institutional quality, and economic freedom indicators, from the Heritage Foundation. Institutional quality shows economic freedom grounded on 12 quantitative and qualitative factors Institutional qualities indicators are divided into four main pillars, namely rule of law (property rights, judicial effectiveness, and government integrity), government size (tax burden, government spending, and fiscal health), regulatory efficiency (business freedom, labor freedom, and monetary freedom) and market openness with three indicators; trade freedom, investment freedom, and financial freedom [13,14]. Thirdly, to the bets of our knowledge, this is the first empirical investigation that examined the effect of government revenue, institutional quality (averaged from 12 economic freedom indicators by reliable source Heritage Foundation), and economic growth in SSA countries in this context.

The rest of the paper is organized as follows: Section 2 describes the theoretical and empirical grounds of government revenue, institutional quality, and economic growth. Further, this section reviewed empirical works related to the control variables of the study. Section 3 describes the methodology and model specification used in the paper while section 4 presents the results and discussions of findings. Finally, section 5 concludes and shows the policy implication of the paper.

## 2. Reviews of related literature

### 2.1. Theoretical reviews of literature

The concept of economic growth theory is grounded in several theories. First, the mercantilism theory of economic growth, one of the pre-classical theories, asserts that economic growth is due to the accumulation of gold and silver. This theory believes in the surplus of gold [15]. The second theory is the classical theory of economic growth, the theory marked the birth of other economic growth theories and stresses that growth is the result of specialization and production returns to scale [16,17]. Thirdly, the limitations of classical theory led to the emergence of neoclassical theory. This theory supports the supply-side growth of input factors. This theory argues also that productivity and the size of the workforce are one of the two major sources of economic growth.

Fourthly, to fill the limitations of the previous economic growth theories, endogenous economic growth theory emerged. This theory is one of the contemporary growth theories that support knowledge spillovers, innovations, and human capital of the nations. Our study is based on the nation of this theory as we modeled the link between revenues of government and institutional quality as factors affecting economic growth [18,19]. The final theory of economic growth we reviewed in this paper is the dependency theory, a theory that celebrates the nature of the relationship between developing and developed nations. This theory argues that economic growth is mainly determined by historical relationships among or between nations [20,21].

Regarding theories of government revenue, first, we start by reviewing the cost of service theory of government revenue. The major source of government revenue is tax revenue. This theory suggests that government tax equity and fairness need to be viewed from the angle of the service delivered to the citizens of that nation [4,22]. Second, we introduce the benefit theory of government revenue. The limitations that underlie the cost-of-service theory of taxation led to the birth of the benefit theory of tax revenue. This is the other side of the rendered government services to society [12,23]. Thirdly, the Socio-political theory of government revenue is reviewed within the context of our study. This theory emanates as a reaction to the limitations of the benefit and cost theory of taxation. The socio-political theory of tax revenue underlies the fact that the tax revenue principle should be determined by the objectives of the government. According to this theory, social and political cases need to be considered to decide the amount of tax revenue [24].

Other most important theories of taxation are also there. First, classical taxation theory, grounded on the Adam Smith cannons of taxation, suggests a fiscal role of government revenue. This theory underlines the fact that state revenue needs to be viewed from the fiscal policy angle. Secondly, the well-celebrated theory of government revenue is Keynesian taxation theory following the ideas of Lord J. M. Keynes during the world's great depression. Keynesian theory of taxation asserts that state revenue is a means of economic stabilization. It is a deliberate tool for recovering from the hit of world great depression [25,26]. Thirdly, we come across Neo-classical taxation theory, the reaction against the classical theory of taxation which asserts that taxation is the means of confronting problems. According to the classical theory of taxation, taxation should be used as a means of correcting market failure and market instabilities. The final theory reviewed due to its relevance to this paper is the Neo-Keynesian taxation theory, the theory by followers of Keynes. This theory suggests that state revenue should be used with a focus on supply-side policies instead of demand-side policies [27,28].

## 2.2. Empirical literature reviews on government revenue and economic growth nexus

[29] Investigated the link between tax and economic growth of South Asian countries with the help of Difference GMM and found that tax revenue decreases economic growth for the study period of 2002 to 2019 in seventeen Asian countries. Similarly [1], found that negative effect of tax revenue on the economic growth of nine South Asian countries for the study period of 2000 to 2020. Previously, the work of [30,31] found that the existing policy structure of SSA countries matters the tax revenue components and their corresponding impact on economic growth. From the analysis made, the study found that indirect tax tends to be supported by the existing policy structure of SSA countries.

The empirical work of [8] found unidirectional causality between tax revenue and economic growth in the short run for the panel of G7 countries from the study period of 1980 to 2016. From the panel causality test carried out, their study concluded that there is long-run bidirectional causality between tax revenue and economic growth [32]. Found that government revenue is beneficial for the economic growth of China in the context of the Laffer curve.

[33] Found that tax revenue harms economic growth for the study period of 2005 to 2020 for 12 regional countries of SSA. Further, their study revealed that their tax revenue is expanding dead weight loss in the region. Their study concluded that African countries should implement either tax cuts or work on tax base expansion. Contrarily [34], found that tax revenue has a significant positive effect on the economic growth of a panel of five West Africans during the study period of 2000–2015.

The study by [35] found that the effect of government revenue on the economic growth of OECD countries is dependent on tax and fiscal policy packages. Tax revenue negatively

contributes to GDP when fiscal policy is applied with negative packages while it is found to enhance economic growth when fiscal policy is applied with positive packages. Their study concluded that policymakers should deliver a positive benefit package of fiscal policy in the OECD region. [36] found that tax harmed the economic growth of 21 OECD countries during the study periods of 965 to 2010 from Sami Parametric techniques of estimation.

The study by [37] found that public revenue positively impacts growth in the long run while this is not the case in the short run; the effect of public revenue is insignificant in the short run. From the panel ARDL model applied for 20 Asian countries, their study concluded that public revenue deters economic growth in both periods. Another similar study by [6] found that income taxes are found to be negative and insignificant for economic growth in developing countries taking Tanzania into focus.

The empirical investigation by [38] revealed that tax revenue is positively linked to the GDP of Africa over the study period of 2004 to 2013. Their study supported the taxation theory of Ibn Khaldun on taxation which promotes high and weak levels of taxation to enhance economic growth in SSA African countries [39]. Found that the effect of tax revenue on economic growth is dependent on its composition and the level of political stability of countries using tax revenue from 2001 to 2018 for seven countries in South Asia. Contrary to this, their study found that custom duties positively and significantly contributed to the economic growth of South Asia.

[40] Found that taxation has zero effect of taxation in the long run for the economy of 32 SSA countries from the period 1980 to 2010. From the panel error correction model utilized in the study, it was concluded that SSA countries should not use taxation as an instrument of fiscal policy. A similar study was by [41] who found that the effect of tax revenue on economic growth in SSA is dependent on the level of stable macroeconomic policies and environments. From the PVECM model employed in their study, external and internal trade policies and economic integrations are the very crucial factors that impede the link between tax and economic growth during the study period from 1992 to 2015.

[42] Found that indirect tax adversely affected the economic growth of Pakistan during the study period of 1974 to 2010 for the economy of Pakistan. Their study also observed that the short-run coefficient of indirect tax was found to be insignificant. Similar studies by [43] found that the effect of taxation on economic growth is dependent on the nature of the economic shock happening to the economy. Further [39], a similar study found that tax revenue enhanced economic growth during the study periods of 1979 to 2017 for the economy of Pakistan. From the autoregressive distributive lag model utilized in their study, their study concluded that the effect of government revenue on economic growth is dependent on revenue collection in the area.

[30,44] Found unidirectional causal flow from tax revenue to economic growth in Ghana during the study periods of 1986Q1-2014Q4 through the use of Granger causality. Their study supported the fact that tax revenue in Ghana can influence economic growth and recommended the expansion of tax revenue bases in the country [45]. Examined the effect of government expenditure on the economic growth of Ethiopia by the use of time series data and the ARDL model estimation technique. The study found that a better tax collection system has a positive and significant effect on the economic growth of Ethiopia from the period of 1974 to 2014 in both the short and long run. Contrary to this [46], found that government revenue hurt the economic growth of Indonesia during the study period of 1990–2020 using the ECM estimation technique.

[47] Examined the effect of government revenue on the economic growth of Romania using time series data for the study period of 1998–2014 through the VAR estimation technique. The study concluded that government revenue does not significantly influence the

economic growth of the country [48]. Employed the Auto-Regressive Distribution Lag (ARDL) approach to investigate the linkage between tax revenue and economic growth in South Africa. Their study found that tax revenue adversely affected the economic growth of South Africa from the period of 981–2016. Contrary to this [49], found that tax revenue positively and significantly affected the economic growth of Ethiopia from the period of 1980 to 2018 through the estimation techniques of the Autoregressive Distributed Lag (ARDL) approach and vector error correction model.

The recent empirical work of [50] found that tax revenue negatively contributed to the economic growth of Pakistan from the study periods of 1985 to 2021 using the ARDL bound test estimation technique. Their study witnessed that the adverse impact of the tax revenue is in both short and long-run periods. Similarly [51], found that the majority of the literature supports a negative link between taxation and economic growth although much of the works of literature reveal positive and non-linear effects of taxation and economic growth.

## 2.3. Empirical literature reviews on institutional quality and economic growth nexus

The link between institutional quality and economic growth has been studied by several scholars. For instance [7], examined the link between tax revenue and economic growth in low and lower-middle-income countries. Their study concluded that institutional quality enhances economic growth. Using the panel data for the periods of 2005 to 2019, they concluded that institutional quality and tax revenue have been linked with economic growth in the both short run and the long run. Their study concluded that the relationship between institutional quality and economic growth varies across samples in the short run while uniformity is observed in the long run.

[52] Examined the effect of institutional quality and economic growth for the panel of East African countries for the study periods of 1996 to 2015. From the fixed effects and random effects methods of estimation utilized in the study, they found that institutional quality has a positive impact on the economic growth of East African countries. Their study strictly recommended the improvement in institutional quality in East Africa [2,3,53].

Jung (2020) improvement in institutional quality enhances productivity and in turn, improves the economic growth of emerging economies. A similar study by [54] also attested that institutional quality promotes economic growth as the two are highly interdependent. Their study concluded that policies that enhance institutional qualities in the area need to be implemented. Further, [55] found institutional quality in African countries from the period of 1999 to 2017 through the use of the FMOLS estimation technique. Further, their study found that there is bidirectional causality between economic growth and institutional quality.

[56] Observed that institutions enhance credit growth and in turn economic growth in the panel data of 60 countries for the study periods of 2003 and 2017. Similarly [57], examined the effect of institutional quality on economic growth. From the panel data employed in the study, their study concluded that there is a U-shaped pattern link between the quality of institutions and economic growth.

[58] Linked institutions and economic growth following the Solow and Mankiw growth models and concluded that long-run economic growth is dependent on the level of institutional change and evolution. A similar study by [59] found that institutional quality enhances the economic growth of a panel of three East Asian countries from the study periods of 1990 to 2016 using Fully Modified Ordinary Least Squares and Dynamic Ordinary Least Square econometric estimation techniques. The Granger causality test conducted in their study found that unidirectional causality runs from the institutional quality of the countries to the economic growth of the region.

[59] Studied the effect of debt volatility and economic growth in 45 Sub-Saharan African countries with the main objective of whether institutional quality matters or not during the period 1980–2017. Using fixed a two-step instrumental variable generalized method of moment (2SIV-GMM), they found that institutional quality improves the economic growth of the region although it fails to eradicate the adverse effect of aid volatility in SSA countries during the period under investigation. Previously similar work of [60,61] found that the effect of institutional quality on saving in SSA is dependent on the level of economic growth of countries. From panel data employed for the period spanning from 1980 to 2015, their study concluded that institutional quality enhances growth in countries with relatively high per capita income while it impedes economic growth relatively low per capita income of SSA countries.

[62,63] Examined the effect of institutional quality on the economic growth of SSA countries. Employing the panel data from the period through the two-step systems GMM estimation technique for the period of 2006 to 2018, their study concluded that institutional quality varies across regions of SSA [64]. Studied the link between fiscal policy and economic growth in CFA countries for the study period of 1995–2017. Their study revealed that the link between fiscal policy and economic growth is contrary to the Keynesian economic theory. This result contradicts the study of [65], who found that institutional quality has a small positive effect on the economic growth of 21 sub-Saharan African countries between the study periods of 1996 to 2012. From the first difference GMM estimation technique employed in the model, improvement in institutional quality variables was recommended in the sample countries.

Regarding control variables, over the last few decades, the effect of foreign direct investment and economic growth captured the attention of recent empirical findings. For example [66], found that foreign direct investment enhances economic growth in the long run while it is statistically insignificant in the short run. From the panel data of 22 countries in SSA for the study periods of 1988 to 2019, the study concluded that attention should be delivered to attracting foreign direct investment. From the panel GMM method of estimation [67], found a contrary finding; foreign direct investment enhanced growth only when entered into the model with an infrastructure in 46 SSA countries for the study periods spanning from 2003 to 2017.

[68] Examined the effect of foreign direct investment on economic growth. Using panel data estimation techniques, the study found that foreign direct investment deters economic growth in some aspects. Similarly [69], employed the autoregressive distributed lag technique to examine the effect of foreign direct investment on the economic growth of West Africa from 1996–2016. Their study also concluded that political governance, the proxy of institutional quality stimulates the positive effect of foreign direct investment.

[70] Examined the link between foreign direct investment and economic growth in SSA countries. Employing, a generalized method of moment estimation technique for the panel of 25 countries in Sub-Saharan Africa for the period 1980 to 2014, their study found that information technology enhances the positive effect of economic growth on the economic growth of the region. From the panel data analysis carried out [71], examined the effect of foreign direct investment on the economic growth of SSA over the study periods of 1990 to 2016.

Similarly [72], found that foreign direct investment is one of the key drivers of South Asian economies over the sample study periods of 1980–2016.

The effect of trade openness has attracted scholarly attention very recently. Consequently, various studies have been carried out in recent years. For instance [59], found that trade openness positively contributes to economic growth in three East Asian countries from the study periods of 1990 to 2016. A similar finding was reported by the study of [73] who studied the link between trade openness and economic growth in 38 SSA for the study periods of 2004–

2019. From the One-Step difference GMM estimation technique, their study found that trade openness had a positive and significant effect on economic growth. The earlier study by [74] also found that the link between trade openness and economic growth in SSA countries reveals an inverted u-curve which is in line with the Laffer trade curve.

[75] Found a positive effect of institutions on trade openness in 34 SSA countries covering the period 1996 to 2016. From the structural equation modeling employed in the study, it is concluded that institutional quality enhances economic growth. A recent study by [76] found that openness has a positive impact on the long-term economic growth of BRICS countries. Similarly [77], examined the link between trade openness and economic growth in sixteen ECOWAS and fifteen SADC countries using pooled OLS, fixed, and random effects techniques for the period 2006–2017. The result of the study depicts that trade openness impacts the economic growth rate but insignificantly.

[78] Found that openness and economic growth exhibited an indirect relationship in 80 countries for the study period of period 1980 to 2014. From the GMM estimation technique adopted in the model, their study also found that trade openness harms the economic growth rate. The contrary finding was reported by [79] who examined the effect of trade openness on the economic growth of SSA countries 40 SSA countries from 1990 to 2017 using panel vector auto-regression and the System of generalized method of moments and found that trade openness promotes growth in the long term in SSA region.

[80] Found that utilized data over 200 years for both developing and developed countries and found that the effect of the population is dependent on the level of economic growth of countries and concluded that low population in high-income countries is a problem while high population in a low-income country is also another challenge. A similar study by [81–83] study was carried out in SSA countries to find that population growth retards economic growth especially where hunger and inequality is the major challenge.

[84] Studied the link between population growth and miraculous growth in China. Their study found that the slow-growing population of China will create a window of demographic change which invites an infinite shortage of labor supply in the country through generating demographic dividend. This generates better opportunities for economic growth. However [85], found contrary to this finding and revealed that demographic dividend decrease which in turn reduces saving and capital accumulation in the country by limiting the supply of working age population.

[86] Employed fixed effects and random effects panel data modeling techniques over the period 1970 to 2011 to examine the effect of population dynamics on the economic growth of West African countries. Their study concluded that mortality still impedes economic growth in the region while demographic transition is still beneficial in Western African countries. By using difference and system GMM estimation techniques [87], examined the effect of population on the economic growth of Africa over the study periods 1980 to 2015. From a panel of 53 African countries, their study revealed that population growth improves the economic growth of Africa positively and significantly.

[88] Found that the decline in the working-age population remains a challenge in South East Asia. Similarly [89], found that the decline in the working-age population in Asia is reducing the so-called population-bonus principle in the region. The finding of their study also revealed that the direct effect of population on economic growth arises from the working-age population while the indirect effect originates from the contribution of the entire population to the saving rate. Adopting the panel co-integration [90], examined the effect of 1980 to 2015 in South Asia and reported a similar finding; population growth has no significant impact on the economic growth of South Asia.

## 3. Methodology of the study

### 3.1. Theoretical framework of the method

Government revenue and institutional quality are one of the major determinants of economic growth. Thus, establishing the theoretical link between the variables is essential here. For example [91], designed a methodological framework in which the link between institutional structure, corruption, and economic growth was investigated. Their study investigated whether institutional quality matters regarding the linkage between corruption and economic growth. The Earlier work of [92] proposed the theory of capital circulation in studying the effect of institutional factors on developing economies. Their work argued regarding the causes of the disparities of regional economic growth. Their work further argued that uneven economic growth is characterized by the regional economic growth of the region [93].

The other study that introduced institutional quality in the economic growth model is [94], which found that institutional quality matters for growth although the development stages of a country are also one of the major determinants of the quality of policy concept. On the other hand [95], found that institutional quality is one of the major determinants of economic growth and levels of income of the country determine the effect of institutional quality. [73] also introduced institutional quality in the model of economic growth that investigated the relationship between trade openness and economic growth [96]. Also introduced the interaction of institutional quality and COVID-19 on the performance of firms in OECD member countries.

Meanwhile, the link between government revenue and economic growth was also framed by several scholars in economic literature. For instance [11], introduced a model in which government revenue and expenditure interact for a panel of seventeen developing and emerging South Asian countries using different GMMs. The earlier work of [8] also modeled the interaction of government revenue and expenditure for G-7 countries through the frequency domain approach of econometrics. [7] modeled two fiscal policy variables (government tax revenue and government expenditure) along with institutional quality in their study.

### 3.2. Type and sources of data

This study employed panel data from 43 SSA countries from the period of 2012 to 2022 (see Table 1). The sample countries were selected based on data availability. Countries such as

**Table 1. Description, source, and expected signs of the study variables.**

| Variables | Short | Description | Source | Expected sign |
|---|---|---|---|---|
| Real GDP | lnGDPR | Shows annual percentage growth rate of GDP per capita based on constant local currency. It is gross domestic product divided by midyear population. Measured as GDP per capita growth (annual %) | WDI | Dependent variable |
| Government revenue | lnGOR | Tax revenue refers to compulsory transfers to the central government for public purposes. Measured as tax revenue (% of GDP). | WDI | +/- |
| Foreign Direct Investment | lnFDI | Foreign direct investment is the net inflow of investment. Measured as foreign direct investment, net inflows (% of GDP). | WDI | + |
| Population | lnPOP | Population shows all residents regardless of legal status or citizenship and values shown are midyear estimates. Measured as Population, total | WDI | +/- |
| Openness to trade | lnOp | Measured as trade (% of GDP). It is the sum of exports and imports of goods and services measured as a share of gross domestic product. | WDI | +/- |
| Institutional quality | lnIQT | Institutional quality represents economic freedom based on 12 quantitative and qualitative indicators, divided into four pillars, namely the efficiency of the regulatory, the size of government, the rule of law, and the open market. | HEFI | +/- |

**Source:** Author's building, 2023, HEFI denotes Heritage Economic Freedom Index.

Eritrea, Malawi, Liberia, Sao Tome Principe, and South Sudan were excluded from this study due to the problem associated with the availability of data. This study included a larger sample size as 43 out of 48 SSA are included in the study. This paves the way for the generalizability of the result of the study to SSA countries. The data for real GDP growth rate per capita (GDPR), population (POP), foreign direct investment (FDI), and openness to trade (OP) is extracted from the world development indicator of the World Bank database [97,98].

On the other hand, the data for institutional quality data were obtained from the Heritage Economic Freedom Index (HEFI). Institutional quality epitomizes economic freedom grounded on 12 quantitative and qualitative factors such as property rights, judicial effective-ness, government integrity, tax burden, government spending, fiscal health, business freedom, labor freedom, monetary freedom, trade freedom, investment freedom, and financial freedom. According to HEFI, institutional qualities are divided into four main pillars, namely rule of law (property rights, judicial effectiveness, and government integrity), government size (tax burden, government spending, and fiscal health), regulatory efficiency (business freedom, labor freedom, and monetary freedom) and market openness with three indicators; trade free-dom, investment freedom, and financial freedom[13,14,99]. Both sources of data employed in this study are dependable data sources and thus data used in this study are highly reliable. A detailed description of the study variables is provided in Table 1.

Table 1 presents the variables of the study and their description. As it is depicted, the study has three control variables; population, foreign direct investment, and openness to trade while there are two major variables of interest in the study; government revenue measured by tax rev-enue as a percentage of GDP and institutional quality measured by economic freedom based on 12 different economic freedom indicators, divided into four pillars, namely the efficiency of the regulatory, the size of government, the rule of law, and the open market. Finally, the explained variable of the study is real GDP which is measured as GDP per capita growth (annual %). Gen-erally, the current article has six variables; five independent and one independent variable.

Table 2 presents sample countries included in the study. Accordingly, 43 SSA countries are included in the study. The reason for the inclusion and exclusion of the sample countries in this study was just the availability of data.

**Table 2. List of Sample countries selected for the study.**

| S.No. | Sample countries | S.No | Sample countries | S.No | Sample countries |
|---|---|---|---|---|---|
| 1 | Angola | 15 | Eswatini | 29 | Namibia |
| 2 | Benin | 16 | Ethiopia | 30 | Niger |
| 3 | Botswana | 17 | Gabon | 31 | Nigeria |
| 4 | Burkina Faso | 18 | Gambia, The | 32 | Rwanda |
| 5 | Burundi | 19 | Ghana | 33 | Senegal |
| 6 | Cabo Verde | 20 | Guinea | 34 | Seychelles |
| 7 | Cameroon | 21 | Guinea-Bissau | 35 | Sierra Leone |
| 8 | Central African Republic | 22 | Kenya | 36 | Somalia |
| 9 | Chad | 23 | Lesotho | 37 | South Africa |
| 10 | Comoros | 24 | Madagascar | 38 | Sudan |
| 11 | Congo, Dem. Rep. | 25 | Mali | 39 | Tanzania |
| 12 | Congo, Rep. | 26 | Mauritania | 40 | Togo |
| 13 | Cote d'Ivoire | 27 | Mauritius | 41 | Uganda |
| 14 | Equatorial Guinea | 28 | Mozambique | 42 | Zambia |
| | | | | 43 | Zimbabwe |

Source: World Bank, 2022.

### 3.3. Model specification

In the first step, the following two dynamic panel data models are estimated to evaluate the effect of government revenue and institutional quality on the economic growth of SSA countries. The functional form of the model is:

$$GDPR_{it} = f(GOR_{it} \ FDI_{it} \ POP_{it} \ OP_{it} \ IQT_{it}) \qquad\qquad 1$$

Where, $GDPR_{it}$ is the annual real GDP growth rate per capita, $GOR_{it}$ government revenue, $FDI_{it}$ is foreign direct investment, $POP_{it}$ is population, $OP_{it}$ is openness to trade, $IQT_{it}$, and represents the institutional quality of SSA countries. *it* show the number of cross sections and time of the panel data respectively.

Eq 1 explains economic growth as an explanatory variable of the study. Five explanatory variables are selected in the study in which government revenue and institutional quality are included.

The first model estimated is:

$$GDPR_{it} = \beta GDPR_{it-1} + \alpha GOR_{it} + X\Upsilon_{it} + \nu_{it} \qquad\qquad 2$$

Where, $X\Upsilon_{it}$ represents the set of control variables such as foreign direct investment, population, and openness to trade, $\beta GDPR_{it-1}$ is the lagged value of the annual real GDP growth rate per capita of SSA countries, $X\Upsilon_{it}$ is the matrix of control variables and $\nu_{it}$ is the random error term. All others are interpreted in Eq 1.

Eq 2 attempts to capture the effect of the variable of interest, government revenue, on the economic growth of SSA countries.

The second model estimated is:

$$GDPR_{it} = \beta GDPR_{it-1} + \alpha GOR_{it} + \Omega IQT_{it} + X\Upsilon_{it} + V_{it} \qquad\qquad 3$$

Where, $\beta$, $\alpha$, $\Omega$ are the coefficients of the lagged value of the annual real GDP growth rate per capita of SSA countries, government revenue, and institutional quality respectively

Eq 3 introduced government revenue and institutional quality in a single model along with the set of control variables.

The final model estimated in the study is:

$$GDPR_{it} = \beta_0 + \beta GDPR_{it-1} + \alpha GOR_{it} + \Omega IQT_{it} + \sigma POP_{it} + \psi FDI_{it} + \varrho OP_{it} + \phi(GOR_{it}*IQT_{it}) + \varsigma_{it} \qquad 4$$

Where, $\varsigma_{it}$ is $\mu_{it} + \nu_{it}$, $\phi (GOR_{it}*IQT_{it})$ show the interaction term between government revenue and institutional quality. All others are defined above.

Eq 4 introduced how the interaction of government revenue and institutional quality affects the economic growth of SSA countries.

Finally, the transformed model of the study is provided as:

$$lnGDPR_{it} = \beta_0 + \beta lnGDPR_{it-1} + \alpha lnGOR_{it} + \Omega lnIQT_{it} + \sigma POP_{it} + \psi lnFDI_{it} + \varrho lnOP_{it} + \phi \ln (GOR_{it}*IQT_{it}) + \varsigma_{it} \qquad 5$$

Where $ln$ is the natural logarithm of the variables of the study.

Eq 5 presents the transformed model and the natural logarithm is included for convenience in interpreting the results of the study. With the introduction of natural logarithm, we maintain elasticity interpretation which is better compared to others.

The variables of the study are adopted from various similar studies [57,78,95]. Following the work of [95] foreign direct investment, population, and trade openness were incorporated into the model. Adopting the empirical work of [100], we used trade volume as the proxy of

trade openness in our dynamic panel data model. By adopting the empirical study of [55], institutional quality in our estimated model denotes economic freedom based on 12 quantitative and qualitative factors, separated into four pillars, specifically the efficiency of the regulations, the size of government, the rule of law, and the open market. Following the recent empirical modeling of [11] the government revenue is measured by tax revenue as a percentage of GDP.

## 3.4. Estimation technique and justification of system GMM

The main goal of this paper is to test whether institutional quality matters on the linkage between government revenue and economic growth in SSA Countries using the panel data from 2012 to 2022. System GMM is the model that best fits with short time (T) and large cross-section (G). Our data best suits the condition of the system GMM. The dynamic panel data is estimated to control for the potential endogeneity of government revenue and other variables (control and explanatory variables). Our study estimated the models (Eqs 1, 2, 3, 4, and 5 above) through the use of [101] to tackle the problems linked with dynamic panel data estimation. For instance, this method of estimation takes into account the occurrence of endogeneity problems that arise from the feedback relationship between government revenue and economic growth in the context of our study.

Further, Arellano and Bond's method of dynamic panel data estimation considers and resolves the correlation problems related to country-specific in our cross-sections such as geography and demographics of the panels. Furthermore, this method of estimation is dubbed superior over other dynamic panel data estimation due to its capacity to combat the autocorrelation problem that arises as a result of the inclusion of lag of dependent variable (lag of economic growth in our context). Finally, our study preferred the [101] estimation technique as it uses only valid instruments in the model. This capacity of the estimation technique is generated from the fact that it uses both exogenous variables and the lag levels of endogenous variables as a means of validating instruments in the specified model.

However, any estimation technique is not without a limitation. Accordingly, the weak instruments remain a challenge in an Arellano and Bond estimation technique. This is because the lagged variables can sometimes generate poor instruments of the first difference equation as witnessed in the empirical study by [102]. As a result of this, we employed [103], who proposed a two-step system GMM as one of the most efficient estimation techniques, to eliminate the problem of weak instrumentation in Arellano and Bond.

Another superiority of the two-step system GMM (2SY-GMM) over the two-stage least squares (2SLS) method is that the former uses both level equations and the difference equation to avoid the concern of weak instrumentation while the latter uses only difference equations. Further, in the former case, there is no threat of weak instrumentation while weak instrumentation can be the major agenda due to level equation in the latter. The other benefit of the two-step system GMM estimation procedure that forced its selection in our study is its advantage in correcting the entire model for the heteroscedasticity problem.

Our study also tests to ensure the efficiency and consistency of the two-step system-GMM estimates. The major reason why a test is adopted in our study is to check for suitable instruments [104]. The further robust check of the estimation was also tested by using the Arellano-Bond test for second-order serial correlations through the AR (2). Both Hansen and Arellano-Bond tests designate the failure to reject the null hypothesis of valid instruments (Hansen test) and the absence of second-order serial correlation (for Arellano-Bond AR (2) test). Failing to reject the null hypothesis of both reflects that the estimated linear model is efficient and consistent, which is a desirable result in a scientific investigation.

**Table 3. Results of descriptive statistics.**

| Statistics | GDPR | GOR | IQT | POP | OP | FDI |
|---|---|---|---|---|---|---|
| Mean | 0.9340 | 15.9722 | 49.9770 | 2.39e+07 | 69.7606 | 3.9107 |
| Maximum | 2.9355 | 20.0289 | 55.9357 | 2.72e+07 | 74.8613 | 6.1939 |
| Minimum | -4.4210 | 14.5758 | 0.0380 | 2.08e+07 | 62.5842 | 2.5645 |
| Standard dev. | 1.8469 | 1.5198 | 15.8233 | 20. 3554 | 3.1527 | 0.9827 |
| Observation | 473 | 473 | 473 | 473 | 473 | 473 |

**Source**: Author's computation from Satata15.

# 4. Results and discussions

## 4.1. Descriptive statistics results

This section discusses the descriptive result of the study.

Table 3 presents the result of descriptive statistics of the study with a total observation of 473. The observation shows that the combination of the cross-section and time series portion of panel data is sufficient for the application of system GMM. On the other hand, the mean value of real GDP (GDPR) in SSA countries during the period under consideration is found to be 0.9340 with a maximum value of 2.9355 and a minimum value of -4.4210. The larger standard deviations of 1.846919 imply that greater disparity in real GDP across SSA countries. Further, the mean value of government revenue (GOR) is 15.9722 with a standard deviation of 1.5198. This implies that the government revenue as a percentage of GDP in SSA countries varies across SSA countries with the maximum and minimum value of tax revenue at 20.0289 and 14.5758 respectively.

Similarly, the mean value of institutional quality (IQT) as measured by the economic freedom index is found to be 49.9770 along with maximum and minimum values of 55.9357 and 0.0380 respectively. The standard deviation of institutional quality of SSA country is 15.8233 showing that the quality of institutions across SSA countries varies. The other implication of the result is that SSA countries have lower mean values of institutional quality reflecting that economic freedom is repressed in the region as the larger the economic freedom index reflects the better institutional quality and the higher the economic growth rate.

Table 4 presents a list of SSA countries with the highest and lowest government revenue. Accordingly, Ghana ranks first out of five countries with the highest government revenue in SSA followed by Senegal, Mauritania, South Africa, and Congo Republic respectively. The mean value of the government revenue as % of GDP is found to be 20.0289 implying that the share of government revenue to their real GDP growth rate is better compared to the other. Contrary to this Burkina Faso stands first out of countries with the lowest government revenue

**Table 4. List of the first five countries with highest and lowest government revenue (2012–2022).**

| Rank | Countries with the highest government revenue | Mean Value of the government revenue as % of GDP | Rank | Countries with the lowest government revenue as % of GDP | Mean Value of the government revenue as % of GDP |
|---|---|---|---|---|---|
| 1 | Ghana | | 1 | Burkina Faso | |
| 2 | Senegal | 20.0289 | 2 | Mauritius | 14.5758 |
| 3 | Mauritania | | 3 | Equatorial Guinea | |
| 4 | South Africa | | 4 | Central African Republic | |
| 5 | Congo, Rep. | | 5 | Chad | |

**Source**: Author's computation from Satata15.

as% GDP while Mauritius, Equatorial Guinea, Central African Republic, and Chad take the rank of second to fifth. This result depicts that the government tax revenue is very low in these countries compared to countries with the highest government revenue.

Fig 1 presents the growth rate of the real GDP of SSA countries over the study period. The graph depicted that the real GDP growth rate of SSA countries started substantially declining in 2019 and reached a lower growth rate in 2020. The significant decline in the real GDP growth rate of SSA countries is associated with the COVID-19 pandemic that hit the entire global economy. After the year 2020, the real GDP growth rate started rising implying that the world is recovering from the deadly shock due to the pandemic.

Fig 2 presents government revenue as %of GDP in SSA countries over the study period of 2012–2022 and the results show that the share of tax revenue as a percentage of GDP is very low and slowly declining before the year 2018. The falling portion of government revenue as a percentage of GDP continued until 2022, just due to the COVID-19 pandemic situation posed on the world. After 2020, the share of government revenue as a percentage of GDP started rising following the effort to recover from the noxious pandemic condition.

Fig 3 show that the institutional quality of SSA countries had shown a significant improvement from the year 2012 to 2014. However, the institutional quality remains almost horizontal line (no visible change happened to it) from 2015 to 2022. However, as it is possible to observe from the graph, a little bit downturn has faced the institutional quality of SSA countries. This shows that SSA countries have a lower economic freedom index compared to other regions of the world.

## 4.2. Effect of government revenue on the economic growth of SSA countries

This section discusses the result of the two-step system GMM on the effect of government revenue and institutional quality on the economic growth of SSA (2012–2022).

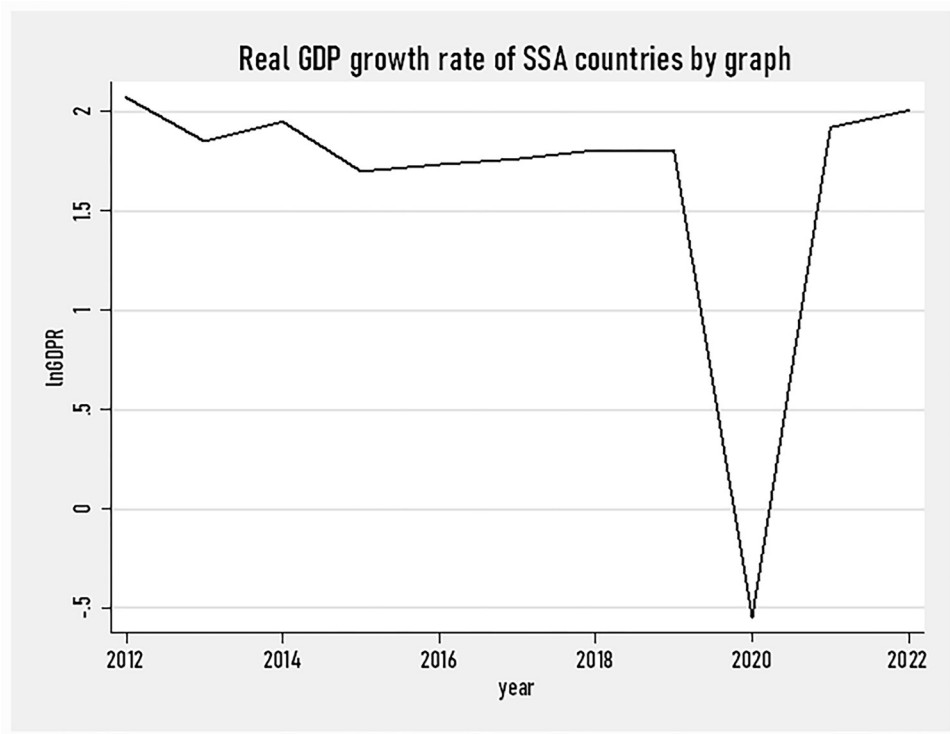

**Fig 1. SSA countries' real GDP growth rate by graph (2012–2022).**

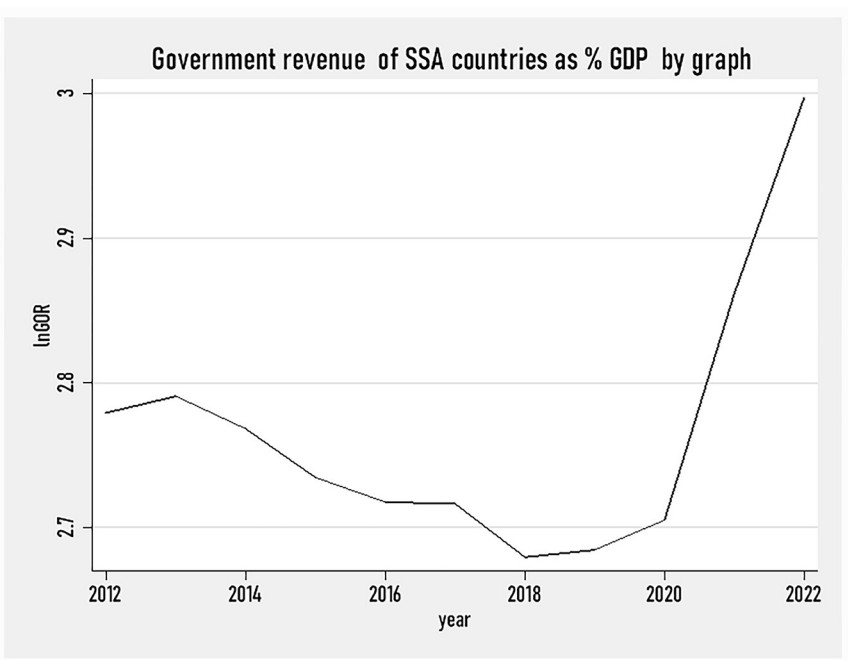

**Fig 2. SSA countries government revenue as % of GDP by graph (2012–2022).**

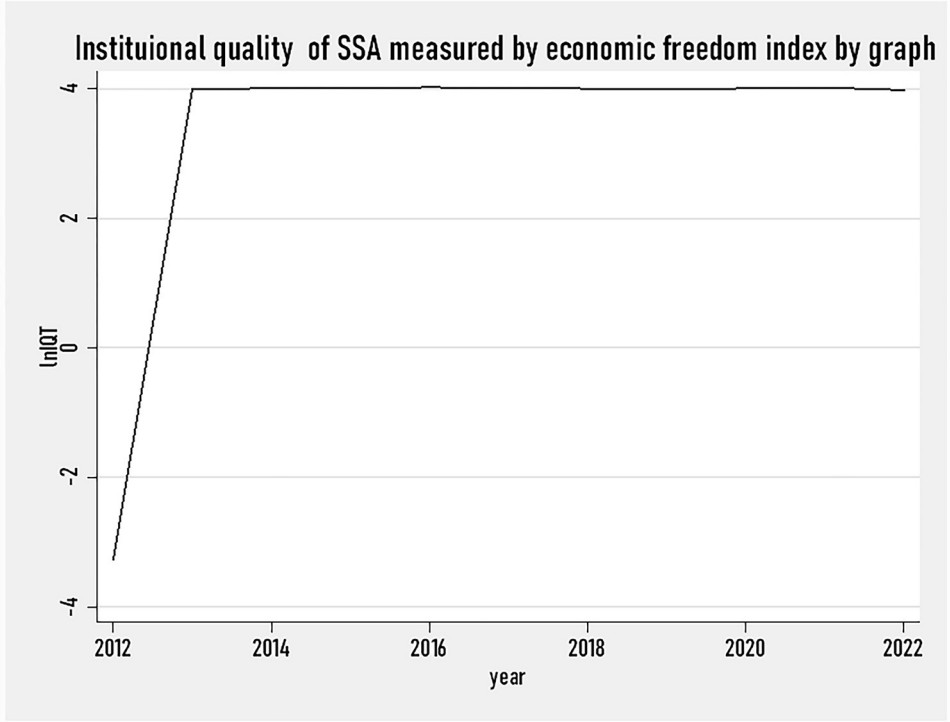

**Fig 3. Institutional quality measured by economic freedom index by graph (2012–2022).**

**Table 5. Results of the two-step system GMM.**

| Variables | Fixed effect(FE) model (1) | | | Random (RE) model(GLS) (2) | | | System GMM(two-step) (3) | | |
|---|---|---|---|---|---|---|---|---|---|
| | Coefficient | t-stat | P>\|t\| | Coefficient | Z-stat | P>\|Z\| | Coefficient | Z-stat | P>\|Z\| |
| L. lnGDPR | 0.3652* | 1.81 | 0.000 | 0.2527** | 2.43 | 0.036 | 0.2600* | 13.05 | 0.000 |
| lnGOR | -0.0826* | -2.12 | 0.000 | -0.0846* | -3.74 | 0.000 | -0.0866* | -15.26 | 0.000 |
| lnIQT | 0.0097* | 1.13 | 0.000 | 0.0158** | 8.28 | 0.034 | 0.02285* | 3.90 | 0.000 |
| lnPOP | -0.0502* | -1.82 | 0.000 | -0.0526* | -11.06 | 0.000 | -0.0898* | -7.31 | 0.007 |
| lnOP | 0.0238* | 2.36 | 0.000 | 0.03896 | 3.82 | 0.000 | 0.0341** | 3.54 | 0.011 |
| lnFDI | 0.3150** | 2.00 | 0.046 | 0.3195** | 2.18 | 0.029 | 0.5476*** | 6.93 | 0.075 |
| Ln(GOR*IQT) | 0.0835 * | 2.91 | 0.004 | 0.0881* | 9.30 | 0.000 | 0.2329* | 23.42 | 0.000 |
| Constant | 19.9719* | 10.95 | 0.000 | 21.8476* | 14.42 | 0.000 | 27.7479* | 17.63 | 0.000 |
| Robustness check of the system GMM | | | | | | | | | |
| R-square | 0.8535 | | | | | | | | |
| Wald test (P-value) | 0.0000 | | | 0.0000 | | | | | |
| AR2 (p-value) | | | | | | | 0.782 | | |
| AR(1) (p-value) | | | | | | | 0.191 | | |
| Hansen (p-value) | | | | | | | 0.195 | | |
| No of instruments | 18 | | | 19 | | | 20 | | |
| No groups | 43 | | | 43 | | | 43 | | |
| Observation | 473 | | | 473 | | | 473 | | |

Source: Author's computation from STATA 15. This table presents the System GMM estimation results for Eq 3 and 4.

*, **, and *** denotes significance at 1%, 5%, and 10% respectively. Refer to Table 1 for definitions of the study variables.

Table 5 presents the result of the model that is estimated from the two-step system GMM. To check the consistency of our estimated results, we estimated the model of the study (Eqs 3 and 4) using other two models; the fixed effect (FE) model (1) and the Random (RE) model (GLS) (2). Our result from the GMM estimation technique was found to be consistent across the three models implying that the result of the study is consistent.

Table 5 further presented that lag one of the real GDP growth rate (L. lnGDPR) from the two-step system GMM is 0.2600, and significant at a 1% level of significance. The result of the system GMM revealed that a one percent increase in the L. lnGDPR in SSA countries is associated with a 0.26 percent rise in the real GDP per capita growth; all other things remain fixed, during the study period. The significance of the result remained consistent across the three models. This implies that the past period's economy positively and significantly contributes to the current period's real growth of SSA over the study period from 2012–2022. Further, the result is consistent with the fact that last year's fertile ground creates a better potential for the current year's economic condition, and real GDP growth rate.

Contrary to this, the two-step system GMM result revealed that the effect of government revenue (lnGOR) on the economic growth of SSA over the study period is negative (-0.0866) and significant at a 1% level of significance before interacting with institutional quality. The finding of our study revealed that a one percent rise in government revenue in the absence of institutional quality, and other factors remain constant, is linked with a 0.08 percent decline in the real GDP growth rate of SSA countries during the sample study period 2012–2022. This result shows that government revenue hurts real growth where institutional quality deteriorates. This finding is consistent with [6,9] but conflicts with the findings of [34], who dealt with only the panel of West African countries, a small cross-section but long time series.

The result of the study further revealed that institutional quality (lnIQT) enhances the growth of SSA. The coefficient institutional quality obtained from the system GMM estimation is positive (0.02285) and significant at a 1% level of significance. From the result we find that a one percent improvement in institutional quality is associated with a 0.02 percent increase in the real GDP growth rate of SSA countries, all other factors remain constant. This implies that institutional quality (economic freedom) is essential for real GDP growth in SSA. This result of our study is consistent with the work of [56,65].

Another variable of interest in this study is the effect of the interaction of government revenue (lnGOR) with institutional quality (lnIQT). The result of the system GMM revealed that although the government revenue adversely impacted the real GDP growth rate of SSA countries before interacting with institutional quality variables, it was found to be positively and significantly enhancing the real GDP growth rate during the period under investigation. The interactive coefficient of government revenue with institutional quality (Ln (GOR*IQT)) is 0.2329 implying that a one percent rise in government revenue in the presence of institutional quality (lnIQT) is associated with a 0.232 percent increase in the real GDP growth rate of SSA countries during the study period under consideration. This result reveals that government revenue enhances economic growth better in the presence of institutional quality. This shows that institutional quality needs to be improved and given attention in Ethiopia. Further, the finding of the study revealed that institutional quality, and economic freedom in the context of our study, matter on the effect of the government revenue on government expenditure. The result also indicates that government revenue, of course, one of the fiscal policy tools, is essential for the real GDP growth of SSA countries when combined with better institutions. The result of our study is consistent with the empirical findings of [7].

Regarding control variables, the result of our study indicated population growth rate (lnPOP) adversely and significantly affects the economic growth of SSA countries. The two-step system GMM result coefficient of lnPOP negative -0.0898 depicts that a one percent rise in population growth rate results in a 0.089 decline in the real GDP growth rate of SSA countries over the period from 2012–2022, keeping all other things constant. This result implies that Africa should control the growth rate of the population to promote growth. This result of our study is consistent with the empirical findings of [81].

In contrast to this, trade openness (lnOP), measured by the volume of trade, is found to be one of the growth-enhancing factors in the panel of sample SSA countries. The coefficient of this variable is 0.0341 which is significant at a 5% level of significance. This implies that a one percentage increase in the volume of trade as a percent of GDP is associated with a 0.034 percent rise in the real GDP growth rate of SSA countries; other things remain constant, during the study period under consideration. This finding is consistent with previous empirical studies of [73,75].

The other control variable, foreign direct investment (lnFDI) measured as foreign direct investment inflow as % of GDP, is one of the factors that enhance the growth of SSA countries. The coefficient of lnFDI, 0.5476, shows that foreign direct investment positively and significantly enhances the growth of the region over the study period of 2012–2022. The system GMM result indicated that other things remaining constant, a one percent increase in foreign direct investment is associated with a 0.547 percent increase in the economic growth of SSA countries during the sample study periods of 2012–2022. This means that the foreign direct investment inflow is a critical variable in SSA countries. This result of our study is consistent with [70,71].

## 4.3. Robustness check of the system GMM model

Table 5, at its bottom section, presents the estimated result of the two-step system GMM, which is employed in the study due to its more robustness in treating heteroskedasticity and

autocorrelation. The result of the Arellano–Bond, AR (2) test, is found with a p-value of 0.782 which is insignificant showing that the model is free from the threat of second-order serial correlation. This reflects that our model is safe in this way. It further assures that there is no endogeneity problem in the estimated model.

On the other hand, the result of AR (1) is observed with a probability of 0.191 which again is insignificant and shows that the model of our study has not suffered from the problem of first-order serial correlation. This shows that our model is safe. Further, the Hansen test of instrumental validity in our study revealed the value of 0.195 which is the desired value reflecting that the null hypothesis of instrumental validity cannot be rejected at a 5% percent level of significance. This implies that the model has passed the examination from the instrumental validity point of view; all instruments utilized in our model are valid. The other robustness check of our two-step system GMM goes to the number of instruments (20) and the number of groups (43). Since the number of instruments is less than the number of groups which is desired and acceptable. Thus, our model passed the entire test.

## 5. Conclusions

This study investigated the effect of government revenue–institutional quality interaction on the economic growth of SSA countries using panel data of 43 SSA countries from the period 2012–2022. Our study employed the system GMM since it treats the problem of first and second-order serial correlation when the lag of the dependent variable is used as an explanatory variable in the model. The panel data sourced from the World Development Indicator and Heritage Economic Freedom Index were examined for unit root tests to avoid the threat of bogus regression.

The result of our study revealed that government revenue has a negative and significant effect on the economic growth of SSA countries before interacting with institutional quality variables. The estimated result of the two-step system GMM has indicated that all other things remaining constant, a one percent increase in government revenue(lnGOR) of SSA countries is associated with a 0.08 percent decline in the real GDP growth rate of SSA countries. However, when the government revenue interacts with institutional quality (ln (GOR*IQT), the real GDP of SSA African countries is found to be positively influenced. Thus, our study concludes that institutional quality matters in the relationship between government revenue and economic growth in SSA countries. The policy implication of this result is that SSA countries need to improve the quality of their institutions if the government revenue is to enhance economic growth. Similarly, the estimated result from the system GMM revealed that a one percent improvement in the interaction of government revenue-institutional quality, keeping all other things constant, is associated with a 0.2329 increase in the real GDP growth rate of SSA countries. This shows that when government revenue is combined with institutional quality, its effect on economic growth is positive. The finding of this study is in line with the endogenous growth theory which advocates that growth is affected by internal factors.

On the other hand, our study found that trade openness (lnOP), and foreign direct investment (lnFDI) enhance the economic growth of SSA countries. Contrary to this, the growth rate of population (lnPOP) adversely affects the real GDP growth rate of SSA countries during the period under consideration.

Grounded on the findings, the study provided the following policy implications. Vis-à-vis the government revenue, SSA countries are advised to work hard on creating awareness of the importance of government revenue to economic growth through seminars and various workshops. Several public and private institutions should loud the benefit of taxation on the economic growth of SSA countries. Besides, the means of better appropriate government revenue

management systems should be designed by SSA countries. First, SSA countries need to maintain improved institutional quality and higher government revenue to enhance economic growth. The fact that institutional quality enhances economic growth also implies that government integrity and judicial effectiveness need improvement as a component of the rule of law section of institutional quality in SSA countries. Further, SSA countries need to reduce the tax burden and enhance fiscal health so that government revenue can promote the economic growth of SSA. Likewise, SSA countries need to improve financial, trade, and investment freedoms as indicators of institutional qualities. Finally, policymakers of SSA countries are well suggested to improve institutional quality by providing efficient regulatory qualities, managing the government size, maintaining the rule of law, and promoting an open market.

## 5.1 Recommendation to further study

The effect of government revenue is a very essential section of fiscal policy where academicians are mainly focusing. Future researchers may focus on the effect of government revenue over a longer period by using other models. Moreover, future studies can divide 43 SSA countries into different income groups to investigate the effect of government revenue on the economic growth of SSA countries. Finally, future studies may also look at what happens to the linkage between government revenue and economic growth of SSA countries when government revenue interacts with trade openness, economic integration, and trade liberalizations.

## 5.2 Limitations of the study

The study is not without drawbacks. First, the study dealt only with 43 SSA countries but did not consider the different income groups of SSA countries. Secondly, the study covered the period from 2012–2022 which is a short period appropriate for system GMM. Thirdly, this study viewed the linkage between government revenue and economic growth from the angle of institutional quality point of view alone.

## Supporting information

**S1 Table. Variables of the study and their URL lincence.**
(DOCX)

## Author Contributions

**Conceptualization:** Atnafu Gebremeskel Sore.

**Supervision:** Wondaferahu Mulugeta Demissie, Atnafu Gebremeskel Sore.

**Validation:** Wondaferahu Mulugeta Demissie, Atnafu Gebremeskel Sore.

**Writing – original draft:** Isubalew Daba Ayana.

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
