## [Decision Letter · Decision Letter 0]

1 Oct 2023

PONE-D-23-27778Effect of government revenue on economic growth of Sub Saharan Africa: Does institutional quality matter?PLOS ONE

Dear Dr. Ayana,

Thank you for submitting your manuscript to PLOS ONE. After careful consideration, we feel that it has merit but does not fully meet PLOS ONE’s publication criteria as it currently stands. Therefore, we invite you to submit a revised version of the manuscript that addresses the points raised during the review process.

ACADEMIC EDITOR:     

I have gone through the manuscript and found it fit for the scope of this journal. The issues investigated are very topical, relevant and stand the potential of advancing knowledge in the extant literature. I provide the following comments for the authors to consider in further improving the study.

The opening sentence of the abstract starts with phrase and seems unrelated with what follows. Please, review the sentence. I will advise that a motivating sentence about the significant or issues surrounding the title should open the abstract. In addition, consider at least one novelty of the study in the abstract. The scope of the study in terms of years should be stated.The introduction is well written however, certain basics are lacking in it. I provided a few as follows; (i) the objective of this study is vague and ambiguous. It should be made explicit and concise. The link between government revenues, institutional quality, and economic growth is not well situated. Authors should improve on it.It is important for authors to position the roles or situation analysis of these variables in the SSA. How as government revenue and other key variables trended in the past few years?  (ii) Contributions need to be improved as it does seem some substantial is presented. Authors should carefully provide relevant contributions of their study to the literature. For instance, how does this study relate with certain SDGs and African Vision 2063? Besides, how your study will improve the current situation in SSA should be clearly stated.The section on review of theory should be summarized in two or three paragraphs. There is no need for sectionalizing the theories as done by the author. Besides, since your outcome variable is economic growth, theories that explain it is more important.Methodological contributions of the study in terms of how estimator employed in this study best fits into the analyses should be clearly stated.The literature review section is far from being acceptable. Kindly reduce the sectiosn to the main variables as thus;Research on government revenue and economic growth: merge both 2.3.1 with 2.3.2 and reduce a review of six to eight papers.Research on institutional quality and economic growth.Provide a System GMM model incorporating your variables. Consult Asong Simplice work. He has done quite well on the estimator. I am sure you will gain more insights from his works.You do not need unit root for model estimated with GMM.The analysis on Hausman test is not relevant for your study since system GMM is far better than difference GMM.The discussion of the results should be linked to recent studies.Conclusion can be improved.The policy recommendations are weak in their present form. They can strengthen it by relating the findings to the recommends.Limitation and future research opportunity are missing. They should be considered.Your work suffers greatly from typo and trivial grammatical error. For instance, it was a place you typed (bets instead of best). Please revise your work thorough.Please do well to read recent study on the methodology should adopt.

Overall, the paper is full of potentials that can be evident if the comments above are addressed. Alternatively, if your revised version fails to address the key issues, I will not hesitate to reject your paper. I encourage you to work harder in improving the manuscript. All the best.

Please submit your revised manuscript by Nov 15 2023 11:59PM. If you will need more time than this to complete your revisions, please reply to this message or contact the journal office at plosone@plos.org. Please include the following items when submitting your revised manuscript:A rebuttal letter that responds to each point raised by the academic editor and reviewer(s). You should upload this letter as a separate file labeled 'Response to Reviewers'.A marked-up copy of your manuscript that highlights changes made to the original version. You should upload this as a separate file labeled 'Revised Manuscript with Track Changes'.An unmarked version of your revised paper without tracked changes. You should upload this as a separate file labeled 'Manuscript'.

We look forward to receiving your revised manuscript.

Kind regards,

Ridwan Lanre Ibrahim

Academic Editor

PLOS ONE

Reviewers' comments:

Reviewer's Responses to Questions

**Comments to the Author**

1. Is the manuscript technically sound, and do the data support the conclusions?

Reviewer #1: Yes

2. Has the statistical analysis been performed appropriately and rigorously? 

Reviewer #1: Yes

3. Have the authors made all data underlying the findings in their manuscript fully available?

Reviewer #1: Yes

4. Is the manuscript presented in an intelligible fashion and written in standard English?

Reviewer #1: Yes

5. Review Comments to the Author

Reviewer #1: Dear Author,

Congratulations! My comments on your paper are provided below.

The abstract is exceptionally well-written, providing a concise overview of the paper's content. Furthermore, the literature review, theoretical underpinnings, conceptual framework, and empirical review sections are not only robust but also comprehensive, using relevant and recent materials.

The methodology employed aligns with the best practices found in economic and development literature. The reporting of the findings and their subsequent discussion is succinct and meaningful, with clear policy implications that can be drawn from the analysis.

Lastly, the conclusions and associated recommendations are highly valuable, providing actionable insights for both researchers and policymakers in the field. Overall, this paper demonstrates a high level of scholarship and rigor.

Thank you for considering my review, and I look forward to seeing this work published.

6. PLOS authors have the option to publish the peer review history of their article (what does this mean?). If published, this will include your full peer review and any attached files.

Reviewer #1: **Yes: **Dr. Lukman Raimi

---

## [Author Response · Author response to Decision Letter 0]

8 Oct 2023

Date: 3rd October, 2023

TO: PLOS ONE 

Subject: Submission of a revised version of the manuscript PONE-D-23-27778 for evaluation 

Title: Effect of government revenue on economic growth of Sub-Saharan Africa: Does institutional quality matter? 

Manuscript No.: PONE-D-23-27778

*isubalewmsc@gmail.com/ isubalewd@wollegauniversity.edu.et

Dear Reviewers,

Greetings of the day!

We are thankful to the reviewers for taking the time to assess our manuscript, for their careful reading, and for their suggestions and valuable comments which helped us to substantially improve the quality of our paper. In revising the manuscript, we have carefully considered all the raised comments and suggestions. We have attempted to succinctly explain the changes made in reaction to all comments. Our reply to each comment in a point-by-point fashion is given in what follows. 

1. Regarding your concern about the abstract

Following your good comments we have corrected the opening sentence of our abstract. We have highlighted it as follows in our manuscript. 

Following the approval of sustainable development goals at the global level, the link between fiscal policy, institutional quality, and economic growth has attracted special attention in economic literature.

Thank you so much for your comments that improved our paper. 

We have also explained the scope of the study in the abstract as follows.

This study scrutinizes the effect of government revenue-institutional quality interaction on the economic growth of 43 Sub-Saharan African countries for the period of 2012-2022.

Novelty statement is also included. We have written it as follows. 

The novelty of this study emanates from the estimation technique designated and the introduction of revenue-institutional quality into the economic growth model of SSA. The result of the study reveals

2. Regarding your concern about the introduction

The introduction is well written however, certain basics are lacking in it. I provided a few as follows; (i) the objective of this study is vague and ambiguous. It should be made explicit and concise. The link between government revenues, institutional quality, and economic growth is not well situated. The authors should improve on it.

Dear Reviewers, thank you for appreciating the introduction section of our manuscript. 

As we did it under the abstract, we have written the introduction more clearly (see page 3 paragraph 4).

We have improved the way we wrote the link between government revenue, institutional quality, and economic growth (we have highlighted improved paragraphs in line with your comments).

3. Regarding your concern regarding the significance of the study 

 Authors should carefully provide relevant contributions of their study to the literature. For instance, how does this study relate to certain SDGs and African Vision 2063? Besides, how your study will improve the current situation in SSA should be clearly stated. Dear reviewers, thank you so much for your comment that improved our paper. We have included the contribution of our paper from the point of view of SDG and Vision 2063 of Africa in the introduction section of our manuscript. 

4. Regarding your concern regarding the theoretical Literature review section 

In line with your good comments, we have summarized Theoretical literature into only four paragraphs (see pages 4 and 5 of the manuscript). We have highlighted the changes that were made to our manuscript. We appreciate your comment that substantially improved our work. 

5. Regarding your concern regarding the Methodological contributions of the study 

In line with your comment, we have explained why the System GMM estimation technique best fits into the analyses (see page 18, section 3.4). 

6. On your concerns Regarding the Empirical Literature review (question 6 to 8 of reviewers comment)

Following your good comments, we have reduced the empirical review sections to the main variables. We have merged sections and significantly reduced the review page (see sections 2.2 and 2.3 of our revised manuscript or pages 6-12 of the revised manuscript). 

7. Regarding your concern about the model of the study (System GMM model)

Thank you for recommending Asong Simplice’s work for our reference. We have benefited a lot from his work. We have checked all Equations (1-5) in our model including our variables. We have carefully incorporated your comments. 

8. You do not need a unit root for the model estimated with GMM.

Thank you so much for your good comment. We are convinced by your comment and eliminated the section of the unit root test. Since the data is short-time data (only eleven years of data), the unit root (random walk of the variables) is not the major threat. 

9. The analysis on the Hausman test is not relevant for your study since system GMM is far better than difference GMM.

Following your good comments, we have avoided the section of the Hausman test. It is almost a universal fact among economists and other researchers in the area that a two-step system GMM is better and provides a more efficient estimator than difference GMM. Moreover, the fundamental fact that two-step GMM is also more efficient is also another very important fact. Your comments are appreciated. 

10. The discussion of the results should be linked to recent studies.

In line with your comments, we have linked the result of the study with empirical literature. Thank you so much again for your comments. 

11. The conclusion can be improved.

We have read and checked our conclusion. Thank you for providing us a chance to edit and recheck the conclusion of our study. 

12. The policy recommendations are weak in their present form.

Incorporating your comments into the work, we have significantly improved recommendations. We strengthened the policy implication (page 27 paragraphs 2 and 4, page 28, paragraph 1).

13. Limitations and future research opportunities are missing. They should be considered.

In line with your comments, we have included future research and limitations of the study (page 28 before reference).

14. Regarding your concern about typos and trivial grammatical errors

Following your comments, we have carefully read the work and corrected the grammatical errors.

---

## [Editor Report · Decision Letter 1]

20 Oct 2023

Effect of government revenue on economic growth of Sub Saharan Africa: Does institutional quality matter?

PONE-D-23-27778R1

Dear Dr. Ayana

We’re pleased to inform you that your manuscript has been judged scientifically suitable for publication and will be formally accepted for publication once it meets all outstanding technical requirements.

Kind regards,

Ridwan Lanre Ibrahim

Academic Editor

PLOS ONE
---

## [Editor Report · Acceptance letter]

17 Nov 2023

PONE-D-23-27778R1 

Effect of government revenue on economic growth of Sub-Saharan Africa: Does institutional quality matter? 

Dear Dr. Ayana:

I'm pleased to inform you that your manuscript has been deemed suitable for publication in PLOS ONE. Congratulations! Your manuscript is now with our production department. 

Kind regards, 

on behalf of

Professor Ridwan Lanre Ibrahim 

Academic Editor

PLOS ONE